# Insights into the Structure of Rubisco from Dinoflagellates-In Silico Studies

**DOI:** 10.3390/ijms22168524

**Published:** 2021-08-07

**Authors:** Małgorzata Rydzy, Michał Tracz, Andrzej Szczepaniak, Joanna Grzyb

**Affiliations:** Department of Biophysics, Faculty of Biotechnology, University of Wrocław, F. Joliot-Curie 14a Street, 50-383 Wrocław, Poland; malgorzata.rydzy@uwr.edu.pl (M.R.); michal.tracz@uwr.edu.pl (M.T.); andrzej.szczepaniak@uwr.edu.pl (A.S.)

**Keywords:** Rubisco, structure, photosynthesis, dinoflagelates, *Symbiodinium* sp., homohexamer

## Abstract

Ribulose-1,5-bisphosphate carboxylase/oxygenase (Rubisco) is one of the best studied enzymes. It is crucial for photosynthesis, and thus for all of biosphere’s productivity. There are four isoforms of this enzyme, differing by amino acid sequence composition and quaternary structure. However, there is still a group of organisms, dinoflagellates, single-cell eukaryotes, that are confirmed to possess Rubisco, but no successful purification of the enzyme of such origin, and hence a generation of a crystal structure was reported to date. Here, we are using in silico tools to generate the possible structure of Rubisco from a dinoflagellate representative, *Symbiodinium* sp. We selected two templates: Rubisco from *Rhodospirillum rubrum* and *Rhodopseudomonas palustris*. Both enzymes are the so-called form II Rubiscos, but the first is exclusively a homodimer, while the second one forms homo-hexamers. Obtained models show no differences in amino acids crucial for Rubisco activity. The variation was found at two closely located inserts in the C-terminal domain, of which one extends a helix and the other forms a loop. These inserts most probably do not play a direct role in the enzyme’s activity, but may be responsible for interaction with an unknown protein partner, possibly a regulator or a chaperone. Analysis of the possible oligomerization interface indicated that *Symbiodinium* sp. Rubisco most likely forms a trimer of homodimers, not just a homodimer. This hypothesis was empowered by calculation of binding energies. Additionally, we found that the protein of study is significantly richer in cysteine residues, which may be the cause for its activity loss shortly after cell lysis. Furthermore, we evaluated the influence of the loop insert, identified exclusively in the *Symbiodinium* sp. protein, on the functionality of the recombinantly expressed *R. rubrum* Rubisco. All these findings shed new light onto dinoflagellate Rubisco and may help in future obtainment of a native, active enzyme.

## 1. Introduction

Ribulose1,5-bisphosphate carboxylase/oxygenase (Rubisco) is an enzyme employed by plants, algae, cyanobacteria and other autotrophic organisms to incorporate CO_2_ into organic compounds, thus it is one of the key photosynthetic enzymes. Rubisco catalyses carboxylation reaction, during which it assimilates CO_2_ and an oxygenation reaction, in which it oxidizes the substrate. In both reactions, the substrate is ribulose-1,5-bisphosphate (RuBP). Due to the fact that Rubisco’s effectiveness of carboxylation is low, and that it also catalyses the unfavourable reaction of photorespiration, it is considered to be a limiting factor of photosynthesis. Consequently, Rubisco is the obvious target for the increase in agricultural production efficiency, and thus it is one of the best studied enzymes for this application [1]. Rubisco consists of at least two catalytic, large subunits (RbcL), and in some cases, of additional regulatory small subunits. To reach catalytic competence, lysine in the active site of Rubisco must first be carboxylated by a non-substrate CO_2_ molecule, followed by the binding of a Mg^2+^ ion. This process is called carbamylation and serves to position the substrate RuBP for an efficient electrophilic attack by the second CO_2_ molecule that will be fixed in the Calvin-Benson cycle (CBB) cycle upon RuBP binding. The active site is closed via two conformational changes in RbcL: loop 6 in the C-terminal domain of RbcL extends over the bound RuBP trapping it underneath; the C-terminal tail of RbcL then stretches across the subunit and pins down loop 6, closing the active site, which results in a closed conformation of Rubisco. Besides RuBP, Rubisco can also bind other molecules like carboxyarabinitol-1,5-bisphosphate (CABP), which is a tight-binding inhibitor of this enzyme, making the active site of carbamylated or decarbamylated Rubisco adopt a closed conformation, and downregulating Rubisco’s activity [2,3,4].

The variation of Rubisco is great due to the huge diversity of organisms that it was found in. Additionally, the different quaternary structures allow distinguishing four different Rubisco forms. Of four known forms, the dinoflagellates form II is the least studied one. Most papers about this form come from the period of 1972–2003 [5,6,7,8,9,10]. Until today all other Rubiscos have been very well studied, while many questions pertaining to the dinoflagellate enzyme are left unanswered. Rubisco from these organisms shows a set of surprising features. Little is known about its catalytic properties, besides the fact that it is highly unstable, however, possesses a much greater specificity factor (SF, defined as the ratio between CO_2_/O_2_ activity), than other form II Rubiscos [10,11]. It is very important to understand the origin of such high SF, as it may help in improving catalytic properties of other Rubiscos. The dinoflagellate Rubisco has been shown to be a form II type enzyme, a homodimer of RbcL (L2), most likely similar to the one from *Rhodospirillum rubrum,* and is encoded by nuclear-localized genes unlike other known eukaryotic large Rubisco subunits, which are encoded by the plastidic genome. What is more, it is encoded as a triple polyprotein by a diverse gene family that contains introns [6,12]. *Symbiodinium* sp. Rubisco expression is photoperiod regulated, but also dependent on its anemone host [13]. Another outstanding fact is that dinoflagellates, although being aerobic photoautotrophs, have a form II Rubisco. This form of Rubisco originates from anaerobic proteobacteria and has a high affinity for O_2_, and this should lead, under normal circumstances for an aerobic organism, to inefficient CO_2_ assimilation. Since this is not true, we may suppose that dinoflagellates cells pose a mechanism to cope with the O_2_ dilemma, e.g., a local CO_2_ concentrating mechanism (CCM) [7].

This unusual set of features of dinoflagellate Rubisco suggest also unusual evolutionary origin, corresponding to the mysterious evolution of dinoflagellate, with multiple events of endosymbiosis [7]. To further understand it, more data is needed about the enzyme itself.

The main obstacle in obtaining sufficient data is that the dinoflagellate Rubisco is highly unstable. It has been shown that Rubisco from *Symbiodinium* sp. and *A. cartere* lost its activity within 30 min following the cell lysis [10], while higher plant or *R. rubrum* Rubisco is stable for several hours and may be easily isolated [11]. The reason for this venture is not fully understood. It was shown that loss of *Symbiodinium* sp. Rubisco activity was not due to proteolysis or precipitation. The explanation may be the instability of the L2 dimer or of the higher quaternary structure complex [10]. There might be some specific chaperone proteins involved in stabilising the final oligomer, what is suggested by Rubisco assembly scenario present in other organisms [14,15]. Existence of chaperones might be deduced from an organism’s genome homology study. However, such is impossible for the dinoflagellate genomes, since they are enormously large (from 1 to 270 Gb, a size that is one-third to 90-fold the size of the human genome), and they have not been fully sequenced so far. Although surely not depicting the whole picture, some chaperones were indeed identified in the *Symbiodinium* sp. transcriptome [16].

An enzyme’s crystal structure would be helpful in understanding the dinoflagellate Rubisco. No successful effort to solve it was yet carried out, mainly because it is impossible to purify its native form due to the aforementioned. However, tools are available to search for the answers not only in vivo, but also in silico. Such an attempt was successfully used for several proteins, which demonstrated as hard to crystallize [17]. The present paper is an attempt to create a model of a structure of the dinoflagellate Rubisco from *Symbiodinium* sp. by homology modelling. We utilize known solved structures of form II Rubisco as templates. Then, we show similarities and differences, which we use to build an explanation for the unusual features of dinoflagellate Rubisco. In a basic experiment, we also show that one of the identified elements (an insert forming loop, exclusive for dinoflagellates) may influence Rubisco solubility.

## 2. Results and Discussion

### 2.1. Homologues of Form II Rubisco from Rhodospirillum Rubrum among Dinoflagellates

To find the best sequence for further modelling, we used the blastP tool to find homologues of the template *R. rubrum* Rubisco among dinoflagellates. As mentioned already, this protein is broadly accepted as a model form II Rubisco. The highest scoring entries are listed in Table 1.

*Heterocapsa triquetra* showed the highest similarity of amino acid sequence to the *R. rubrum* sequence, as described by Query cover (97%, a number that describes how much of the query sequence is covered by the target sequence), E value (0.0, expected value, a number that describes how many times a match by chance in a database of that size is expected; the lower the E value is, the more significant the match) and percent identity (67.67%, a percent of identical amino acids in the same position of the sequence) [18]. The best studied Rubisco from dinoflagellates is the one from *Symbiodinium* sp., being the second with the highest score. It differs from the first hit by less than 2 in percent identity. Thus, we decided to choose *Symbiodinium* sp. as a case for further investigations in this paper.

### 2.2. Analysis of the Amino Acid Sequence of Dinoflagellate Rubiscos

To compare the primary structure of dinoflagellate Rubisco, we aligned sequences of Rubiscos listed in Table 1 on the *R. rubrum* template using Clustal OMEGA [14]. This comparison showed differences that might be crucial for further investigation of the eukaryotic form II Rubisco (Figure 1A).

First of all, in our alignment dinoflagellate Rubiscos do not start with a methionine residue (like in *R. rubrum*), but with a leucine. The lack of an initial codon suggests that there might be a transit peptide encoded at the beginning of the rbcA locus, which encodes rbcL. Rubiscos from dinoflagellates are encoded in the nucleus, and therefore need to be transported into the chloroplasts. It was previously shown that there is an upstream sequence in the rbcA mRNA, with a pattern of conserved residues analogous to Euglena’s Rubisco’s small subunit precursor polyprotein [6]. Aranda and co-workers sequenced and analysed parts of the dinoflagellate genomes and transcriptomes, and identified this upstream sequence of the rbcA locus [8]. The second reason for the lack of methionine is the protein’s encoding as a precursor polyprotein. This means that first result of translation is a longer peptide, bearing a transit peptide, and two or more proteins, which are separated with spacers. This pre-polyprotein trend occurs also in Euglena’s proteome, where, for example, light-harvesting complexes consist as such, and are separated with a deca-peptide spacer [10].

As mentioned previously, more than 67% of the amino acid sequence is identical in aligned proteins. Most of the differences are equally distributed along the compared sequences. The charge distribution is similar; an isoelectric point of *Symbiodinium* Rubisco is slightly higher than that of *R. rubrum* enzyme (5.72 vs. 5.60). This is a result of a plus one negative and a minus one positive amino acid in the *Symbiodinium* sp. sequence. More notable might be the higher amount of cysteine residues in the dinoflagellate Rubisco. In the *Symbiodinium* sp. sequence, there are 9 such residues, which is almost twice their number (5) in *R. rubrum*. Notably, only two cysteine residues are conserved between *R. rubrum* and dinoflagellate Rubiscos (Cys59 and Cys180). Cysteine residues, although not involved directly in Rubisco activity, are known to be responsible for its redox regulation and conformational changes [3,19]. The importance of cysteine residues was also proven for Arabidopsis thaliana Rubisco; after oxidative inactivation, the enzyme was reactivated by redox treatment [20]. On this basis, we may hypothesise, that the higher content of Cys residues is responsible for possible oxygen-dependent inactivation of *Symbiodinium* sp. Rubisco upon isolation.

The most significant differences between dinoflagellate and *R. Rubrum* Rubiscos are the two insertions present in the dinoflagellate RbcL amino acid sequence (Figure 1A, red rectangles). The first insertion contains three negatively charged amino acids in position 413, and the second insertion is made up of eight amino acids in position 425. Both inserts may be treated as one longer, dinoflagellate-specific motif. The possible role of those inserts will be further discussed on the base of constructed models.

### 2.3. Instability of the Enzyme

High instability of Rubisco from dinoflagellates is the main barrier for further improvement in understanding of this enzyme’s function. We mentioned earlier that the enzyme’s instability is most likely due to conformational issues. It is highly possible that for the folding and assembly of a holoenzyme, some chaperone proteins are needed. Such hypothesis was previously assumed, based on the fact that no precipitation of the protein or proteolysis was observed upon cell lysis [10]. Here, we used the Xtalpred tool to validate whether instability comes from the enzyme’s disordered regions [21,22]. Xtalpred also allows for a calculation of crystallization probability based on the instability index and coil regions. Interestingly, Rubisco from *Symbiodinium* sp. belongs to the low crystallization classes, meaning that the crystallization of this protein might be successful (Table 2). Its instability index is lower than 40, predicting the protein as stable.

### 2.4. A Template Selection for Structure Modelling of Symbiodinium sp. Rubisco Using the SWISS-MODEL Software

There are crystal structures of all known forms of Rubisco including RLPs. The first ever crystal structure of form II Rubisco was the one from *Rhodospirillum rubrum* [23]. Dinoflagellate Rubisco, as mentioned earlier, is also considered to be a form II Rubisco, based on its sequence homology. However, there is no crystal or NMR structure of this enzyme due to its high instability. Thus all that can be done now, is the structure modelling based on homology of known structures. A convenient tool for model prediction is SWISS-MODEL, a fully automated protein structure homology-modelling server [24,25]. As discussed already, we chose the *Symbiodinium* sp. Rubisco sequence for structure modelling. The first step consisted of identifying a proper template based on which homology model was to be built. The algorithms collect templates and list them together with relevant structural information that can be readily used to rank the templates and select the best one according to user-defined criteria. After manual inspection of the obtained results list, we chose the obvious hit of *R. rubrum* for further work. Surprisingly, among suggested templates, the Rubisco from *Rhodopseudomonas palustris* was shown to be the one with the highest similarity and best energetic parameters, and so we also included it as a template. This Rubisco is a unique hexamer with three pairs of catalytic large subunit homodimers around a central 3-fold symmetry axis [26]. Such facts also allowed us to hypothesize that the previous dogma of dinoflagellate Rubisco being a dimer, and not a higher quaternary structure, may not be true. A dimer was postulated on the basis of studies carried out in the 1990s, as well as, in the beginning of the 21st century and was not refined until now. Rubisco from *R. palustris* is an even more suitable template (compare Table 3) than the one from *R. rubrum,* based on the GMQE score (Global Model Quality Estimation). We chose to build the models on these two templates to verify whether *Symbiodinium* sp. Rubisco is a dimer or a hexamer.

### 2.5. Accuracy of the Models of Symbodinium sp. Rubisco Structure Made by Homology Modelling in SWISS-MODEL

We built two models based on two different templates of the Rubisco protein, one from *R. rubrum* (L2 homo-dimer) and one from *R. palustris* (hexamer). Differences in parameters of both models are significant and show that the model built on the Rubisco from *R. palustris* is energetically more favourable. Model template alignment and the structures are presented in colours based on QMEAN model quality (Figure 2). This allows the visualization of regions of a model that are either well or poorly modelled. Local quality is presented in blue and red colours, whereas blue presents a high quality of the modelled region and red shows poor accuracy. This value also represents the “degree of nativeness” of the structural features observed in the model. QMEAN Z-scores around zero indicate good agreement between the model and experimental structures of similar size. Scores of −4.0 or below represent models with low quality. In our case, the model based on the *R. palustris* Rubisco has a −1.13 QMEAN score, and the *R. rubrum* Rubisco based model a −4.37 QMEAN score meaning, that the first one shows the structurally closest model to the original one from *R. palustris* and has the highest quality. The accuracy of models may be a confirmation of an earlier hypothesis that dinoflagellate Rubisco is rather hexameric, in opposition to the previously suggested L2 type homodimer. There is only one poor quality region in our modelled Rubisco; this is the insert region with the peptide FGNISLSD. This insertion is conserved only among dinoflagellate Rubisco, thus there was no template available to model this fragment.

### 2.6. Structure of the Active Site in a Modelled Rubisco from Symbiodinium sp.

There are two X-ray structures of form II Rubisco from *R. palustris* in PDB database: A structure of an activated CABP-bound form II Rubisco (4FL1) and of an activated apoenzyme with two sulphate ions bound (4FL2). For this project, we chose the 4FL1 structure, as we mentioned earlier this one has better parameters in terms of model building. This structure also contains CABP in the active centre. CABP is a naturally occurring sugar phosphate and a tight binding Rubisco inhibitor, causing the active site of carbamylated or decarbamylated enzyme to adopt a closed conformation [2,27]. Thus, the model we built represented an activated, closed conformation (Figure 3C,D). On the other side, based on *R. rubrum* Rubisco we built a model representing an activated, open conformation with a substrate, RuBP, bound in the active site (Figure 3A,B). Comparison of all residues in the active sites of Rubisco from *R. palustris*, *R. rubrum* and our modelled structures of *Symbiodinium* sp. Rubisco showed that there are no significant differences except the open/closed conformation. All conservative amino acid residues of active sites among all forms of Rubisco have been noticed to be in the same positions (see Appendix A, Appendix A).

### 2.7. Analysis of a Possible Role of Insertions in the 413 and 425 Positions

The 413 insert consists of three amino acids (G/D, E, E) and extends a helix by one turn, while the 8-amino acid 425 insert in our model is a loop, exposed to water (Figure 4). The helix and the loop are in the C domain of the large subunit. A carboxyl terminus of Rubisco is the centre of the catalysis and has a unique conformation when is activated, and when it is bound with CABP [2]. However, the discussed inserts’ location excludes direct involvement in catalytic activity of the enzyme, although it does not exclude involvement in regulation of its activity (Figure 2B,D and Figure 4B,C). It does not seem to be involved in a dimerization interface between Rubisco’s monomers, as well as in a oligomerization interface of higher-order oligomers. However, this motif is highly conserved among dinoflagellate Rubisco, suggesting that it plays an important role in these species.

To gain a better insight into a possible function of the 425 insert (the 413 insert is too short for a such procedure), we performed a search using the blastP tool with the *Symbiodinium* sp. insert as a template, to find any homological sequences. The search resulted in only five hits of homological peptides (excluding obvious homology with the dinoflagellate Rubisco), which are listed in Table 4.

Among found peptides there are two eukaryotic ones: a cubilin homologue from *Drosophila willstoni*, and a Heat Shock protein from *Fasciola hepatica*, as well as prokaryotic ones from *Proteobacteria* and *Gemmatimonadetes*. All of the found peptides (except for a hypothetical protein from *Proteobacteria* bacterium of unknown function) are chaperone proteins that contribute to cellular response and ions uptake. This indicates a possible role of the insert in an interaction with an unidentified protein partner and may be therefore responsible for the Rubisco enzyme stabilization in vivo. On this basis, we may also postulate that the short, negatively charged 413 insert is an additional patch for binding with the putative interaction partner.

### 2.8. Oligomerization Interface Analysis

The basic Rubisco functional unit is a homodimer. However, in many cases, such dimers may form higher-level oligomers, which help to pack more molecules in the available space, increasing net CO_2_ assimilation. Formation of an octamer is important for higher plant Rubiscos (form I), as well as the recently described form I’ (lacking the small subunit) from *Anaerolineales* [28]. Interestingly, some residues with potential to improve CO_2_ fixation were identified in the oligomerization interface of *Thermosynechococcus elongatus* Rubisco [29]. In all cases, the oligomerization interface consists of hydrogen bonds and salt bridges.

Until now, *Symbiodinium* sp. Rubisco, and other dinoflagellate Rubiscos, were thought to be just homodimeric. Such conclusion was drawn based on their homology to the *R. rubrum* enzyme, which functions exclusively as a dimer. However, our study indicated that there is a high homology of dinoflagellate Rubisco to the same enzyme of *R. palustris*, shown recently to be a hexamer [30]. An indication of the possibility of a hexamer formation by *Symbiodinium* Rubisco may come from an analysis of the probable oligomerization interface. In Figure 5, we compared the molecular surfaces of Rubisco for *R. rubrum*, *R. palustris* and two models obtained for *Symbiodinium* sp. We found that only Rubisco from *R. rubrum* forms a dimer since the outer surface is mostly acidic, with a small amount of basic and hydrophobic patches. For the *R. palustris* enzyme, there is a clearly marked patch of basic and hydrophobic residues. The basic residues may easily form bridges with acidic ones over at the next dimer, while the hydrophobic strip may help to stabilize the binding, if matched to a similar one over at the partner molecule. Very similar patches are found over the *Symbiodinium* sp. Rubisco surface, indifferent to of the template used for modelling. This finding strengthens the possibility of the enzyme’s hexamerization.

For additional verification, we have calculated the theoretical energies of complex formation using the FoldX suite [31]. First, we identified the putative “between dimers” interfaces. These are created by interactions between monomers B and C (58 residues), D and E (57 residues), and F and A (57 residues). The FoldX output provides detailed parametrization of energy, responsible for each complex in the analysed structure; it also includes the internal dimer interface. In Appendix A, we summarized the binding energy of these two types of interfaces in template structures (*R. rubrum*, *R. palustris*), as well as models of *Symbiodinium* sp. and Δloop mutants of *Symbiodinium* sp. (lacking an insert of the 425 loop) and *R. rubrum* (with the same loop added). For comparison, we also included a model created with *R. rubrum* RbCL sequence on the *R. palustris* structural template. We also listed the electrostatic component of the binding energy, as we hypothesized that this might drive the interface formation. In the first attempt, we found that the energies were affected by a high contribution of the van der Waals clashes component; to avoid such artifacts, prior to the energy calculation, we attempted structure optimization in FoldX.

Of initial notice is the fact that the dimer stability of *Symbiodinium* sp. RbcL was significantly lower (so, binding is tighter), when the modelling template was *R. palustris,* than that of the *R. rubrum* protein structure (−48.93 kcal/moL vs. −34.07 kcal/moL). This is again an indication, that the *R. palustris* RbcL structure was the best template of choice for *Symbiodinium* sp. RbcL. Interestingly, this computational experiment also suggests that the 425 insert does not influence dimer stability of *Symbiodinium* sp. RbcL, but its introduction slightly destabilizes the *R. rubrum* protein.

The binding energy of the interface between dimers (responsible for the RbcL hexamer formation) is generally lower than the energy of dimer binding. For the X-ray confirmed hexamer, RbcL of *R. palustris*, it is on average at −5.86 kcal/moL (particular values are listed in Appendix A). The electrostatic component of the binding energy is at −4.16 kcal/moL. As the opposite, there is the *R. rubrum* protein, as the X-ray confirmed dimer, with the dimer-dimer binding energy of 9.41 kcal/moL (−0.50 kcal/moL of the electrostatic component). Positive binding energy of such high degree indicates that binding is thermodynamically unfavourable. The calculation for *Symbiodinium* sp. provided for a negative dimer-dimer binding energy (−1.15 kcal/moL), although higher, than the one of *R. palustris* protein (−5.86 kcal/moL). In fact, the value for *Symbiodinium* sp. may be even lower, as for one of the interfaces (A to B) the optimization did not eliminate all the clashes.

Intriguingly, the electrostatic component equalled to −3.47 kcal/moL, which is much closer to *R. palustris* than to *R. rubrum*. We may then hypothesize that indeed oligomerization into a hexamer is thermodynamically favourable and is driven by electrostatics.

### 2.9. The Loop of the RbcL from Dinoflagellate Has Measurable Impact on the Enzyme’s Solubility

The novel identified insert 425, which appeared as a loop in the modelled structure, shows poor quality in the terms of energy accuracy. We decided to investigate whether this insert has an impact on solubility of RbcL. For this purpose, we designed two mutants: first with the loop removed from the dinoflagellate RbcL sequence, and a second, with the same loop inserted into RbcL from *R. rubrum* (Figure 6). Then, we assessed the expression and solubility of such RbcL proteins.

The previous studies on RbcL from dinoflagellate suggested that this protein is not expressed in *E. coli* cells due to its high instability [5,10]. Surprisingly, the *Symbiodinium* sp. Rubisco turned out to be expressed in our *E. coli* system.

Figure 7 shows the expression and solubility studies for all four proteins. At first sight, there was no significant difference in the amount of soluble protein in the cell lysate of *E. coli* expressing *Symbiodinium* sp. RbcL comparing to *E. coli* expressing *R. Rubrum* RbcL (Figure 7A,B). Unfortunately antibodies against RbcL form II do not react with the *E. coli* expressed proteins for both *R. rubrum* and the dinoflagellate RbcL in denaturing conditions (after SDS PAGE analysis), so we could not clearly identify and quantify the RbcL bands. Therefore, we turned to Western blotting of a native PAGE gel, which allowed a proper detection (Figure 7B.) As molecule’s native PAGE mobility is not only mass-dependent, and we detected multiple bands, we launched a second direction electrophoresis (black arrows on Figure 7C show, which bands were chosen for the second direction electrophoresis). The molecular masses of Rubiscos from both *R. rubrum* and *Symbiodinium* sp. are expected to be around 51 kDa (as calculated based on amino acid composition), and the second direction electrophoresis produced a band at this level (indicated by the blue arrow on Figure 7C). There is no difference in the amount of protein at the 51 kDa level between *R. rubrum* WT and *R. rubrum* Δloop. The lower band of the native PAGE did not produce a band at the 51 kDa level after analysis by second direction electrophoresis (data not shown), and most probably is not a fully expressed RbcL peptide or its degradation product.

ImageJ densitometry analysis indicated a lower amount of RbcL protein in *Symbiodinium* sp. With the deprived loop, compared to the WT version of the protein. On the other hand, the loop insertion did not change the solubility of the *R. Rubrum* Rubisco. These suggest that the loop is indispensable for *Symbiodinium* sp. RbcL, but has no positive impact on an already well soluble protein.

## 3. Materials and Methods

### 3.1. Sequence Analysis

Sequence of the RbcL from *Rhodospirillum rubrum* [uniport number: P04718], the best studied model of Rubisco form II, was used to search for Rubiscos among dinoflagellates using the blastp tool with default parameters set (Organism—dinoflagellate; taxid:2864) [18]. Next, found sequences of several RbcL from dinoflagellates were aligned using Clustal OMEGA online tool [32]. Resulted RbcL sequences were then aligned to compare form II Rubisco from eukaryotic dinoflagellates and prokaryotes. Finally, we chose *Symbiodinium* sp. Sequence for structure modelling, due to its high level of homology to *R. rubrum*, but also because of the richest set of available literature data amongst dinoflagellate Rubiscos.

### 3.2. Crystallization Prediction

To verify whether, if purified, crystallization of the *Symbiodinium* sp. Rubisco would be feasible we employed the Xtalpred tool for crystallization prediction [21,26,33,34].

### 3.3. Model of the Structure of Rubisco from Symbiodinium sp.

Structure of the *Symbiodinium* sp. was predicted by homology modelling using the SWISS-MODEL tool [24]. The online server was used for all steps of the modelling. Templates selected by the tool in Protein Data Bank (PDB) were manually inspected and two templates with the highest homology to *Symbiodinium* sp. Rubisco were used for modelling.

### 3.4. Computation of Chemical and Physical Parameters

Amino acid sequences of the Rubisco from *Symbiodinium* sp., *R. rubrum* and *R. palustris* were analysed to compare their chemical and physical parameters such as isoelectric point, instability index and aliphatic index using the online tool Protparam [35]. Calculation of energies was done with FoldXsuite 3 [31]. Densitometry analysis was done with ImageJ [36].

### 3.5. Construction of Expression Vectors pUC18RbcLrubrumLoop, pUC18RbcLdinoLoop

pUC18 expression vectors carrying the wild-type codon-optimized RbcL gene coding sequence of *R. rubrum* (GenBank: CAA25080) or *S. microadriaticum* (GenBank: OLP96161) were ordered from Genomed S.A., Warsaw, Poland. The latter was disposed of its chloroplastic signal peptide coding sequence. The *R. rubrum* Δloop and the *S. microadriaticum* Δloop mutants were generated by PCR-based site-directed mutagenesis of the expression vector (loop nucleotide sequence insertion and deletion, respectively), followed with the PCR product phosphorylation by the T4 PNK kinase (Thermofisher, Waltham, MA, USA), and a subsequent ligation by the T4 ligase (Thermofisher, Waltham, MA, USA). DNA sequences of all the resulting constructs used in this study were confirmed by sequencing (Genomed S.A., Warsaw, Poland). Primers for the PCR reaction were as follows in Table 5 (expression vector complement primer sequences are capitalized).

### 3.6. Expression of WT and Mutant Rubisco from R. Rubrum and Symbiodinium in E. coli

Plasmids pUC18RbcLrubrumLoop, pUC18RbcLdinoLoop and plasmids with wild type Rubisco: pUC18RbcLRubrum, pUC18RbcLDino were transformed into the BL21 E. coli strain. Transformed cells were selected on LB-Amp medium (containing 100 μg mL^−1^ of ampicillin). Single colonies were grown in 2 mL LB-Amp liquid medium overnight at 37 °C, and 0.1 mL was used to inoculate 100 mL LB-Amp liquid medium. The cultures were grown at 37 °C to an absorbance at 600 nm of 0.25 before inducing with 1 mM IPTG overnight at 30 °C.

### 3.7. SDS-PAGE, Native-PAGE, Immunoblot Analysis, Protein Quantitation

Proteins were isolated from cells and separated on a Native-PAGE TGX 7.5% gel. Proteins were next blotted onto a PVDF membrane [37] and immunoprobed with anti-RbcL form II antibodies Agrisera^®^ AS15 2955(Gentaur Molecular Products BVBA, Kampenhout, Belgium). Bands chosen after immunoblot analysis where next cut from the gel and used for second direction electrophoresis in denaturing conditions (8% SDS-PAGE) [38]. For all PAGE analyses, the same amount of protein was used. Protein concentration was assayed using ROTI^®^Nanoquant (Carl Roth GmbH, Karlsruhe, Germany), a modification of the Bradford method [39]. 

### 3.8. Chemicals

All used chemicals were pure for analysis. If not stated otherwise, they were purchased from (Carl Roth GmbH, Karlsruhe, Germany). 

## 4. Conclusions

To conclude, we built a structural model of dinoflagellate Rubisco based on known form II homologs of this enzyme. Dinoflagellates, as mentioned, belong to the *Eucaryota*, but their Rubisco, classified as type II, is nuclearly encoded in three repeats, differently to other known eukaryotic Rubiscos of type I. This feature may reflect the evolutional history of the Rubisco enzyme, as dinoflagellate Rubisco shows characteristics of both eukaryotic and prokaryotic organisms. It should be kept in mind that this is an in silico study without crystallographic confirmation; however, it comes out with several indications, which may help in further studies. First, we confirmed that the catalytical site of the enzyme is conserved, and therefore is not an explanation for differences noted between dinoflagellate Rubiscos and its homologs from other organisms. Therefore, the experimentally observed loss of activity of isolated dinoflagellate enzyme must be linked to other structural features of the protein.

We found, that Rubisco from *Symbiodinium* sp. has twice as many cysteine residues as the Rubisco from *R. rubrum*. We postulate that the higher amount of cysteines, which are known to be responsible for redox regulation, might be the cause for high instability of dinoflagellate Rubisco. This observation suggests that the isolation of an active enzyme from a natural source may need additional optimization of redox conditions; the active enzyme expression in a heterological system may also require overcoming of the folding limitations.

Our analysis showed that the dinoflagellatae Rubisco is a hexamer (a trimer of dimers) rather than, as previously suggested, a L2 type enzyme. The indicated hexamer has a more complex structure than a simple dimer. This knowledge might help to obtain a stable purified enzyme, mostly by including chaperone proteins in the process, aiding in formation of a higher oligomer. We may hypothesize that these might be, at least in part, the chaperones alike to those of higher plants; however, it needs further experimental confirmation.

We also show that dinoflagellate Rubiscos contain a novel motif, consisting of a helix extension and a loop. Location of this motif excludes its direct involvement in a catalytical reaction, suggesting rather a role in interaction with an unknown protein partner of possible regulatory function. As a proof of concept, we expressed the *Symbiodinium* sp. RbcL without the loop, finding the protein solubility to be on a significantly lower level. This loop; therefore, maybe important for the interactions with other proteins, such as a possible unknown regulatory protein as well as chaperones. Again, this makes the dinoflagellate enzyme more similar to the eukaryotic Rubisco due to the similar need for a series of chaperone proteins in order to assemble into an active enzyme. All these findings bring us closer to explaining dinoflagellate Rubisco’s surprising features. Full understanding of Rubisco characteristics will make possible reengineering it to gain a higher yield of CO_2_ assimilation, what may benefit in higher crop yield and an overall improvement in biosphere CO_2_ level.

## Figures and Tables

**Figure 1 ijms-22-08524-f001:**
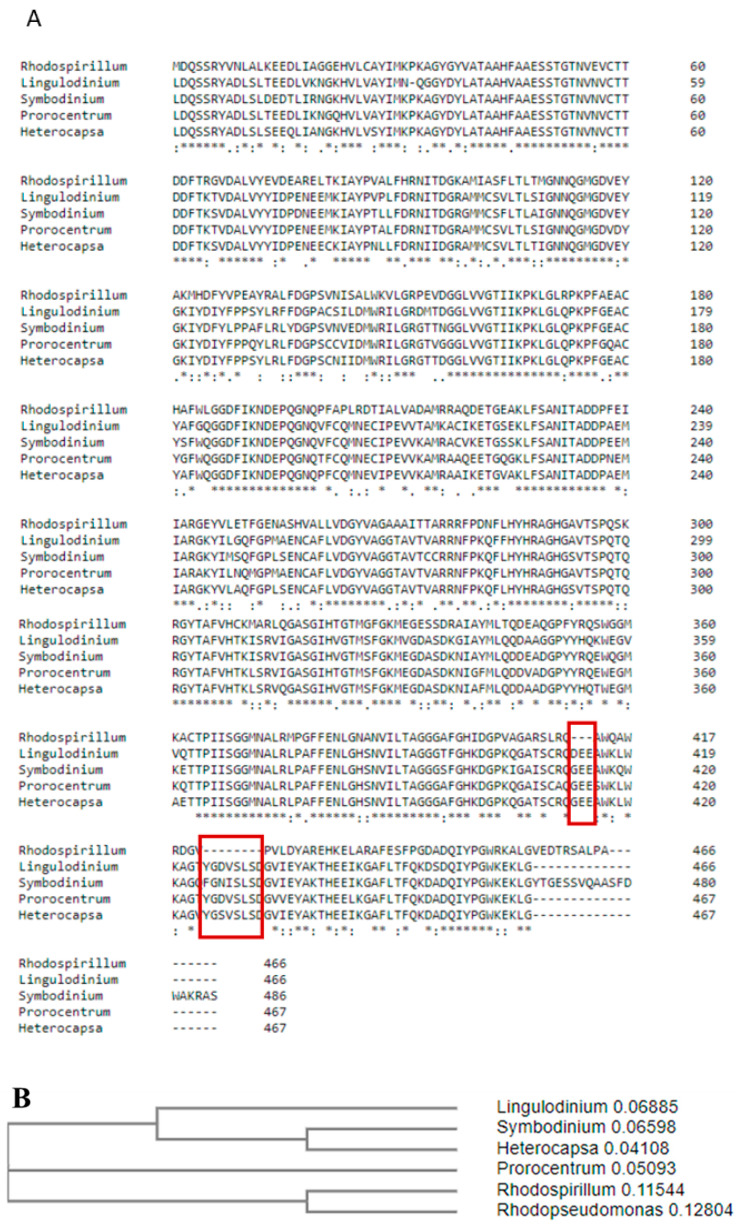
Protein sequence alignment in Clustal OMEGA (**A**) and a phylogenetic tree of form II Rubiscos from Dinoflagellates constructed based on this alignment (**B**). Red frames indicate the position of two unique inserts. “*” indicate identical amino acids in all sequences, “:” indicate amino acids which are not identical but have similar properties.

**Figure 2 ijms-22-08524-f002:**
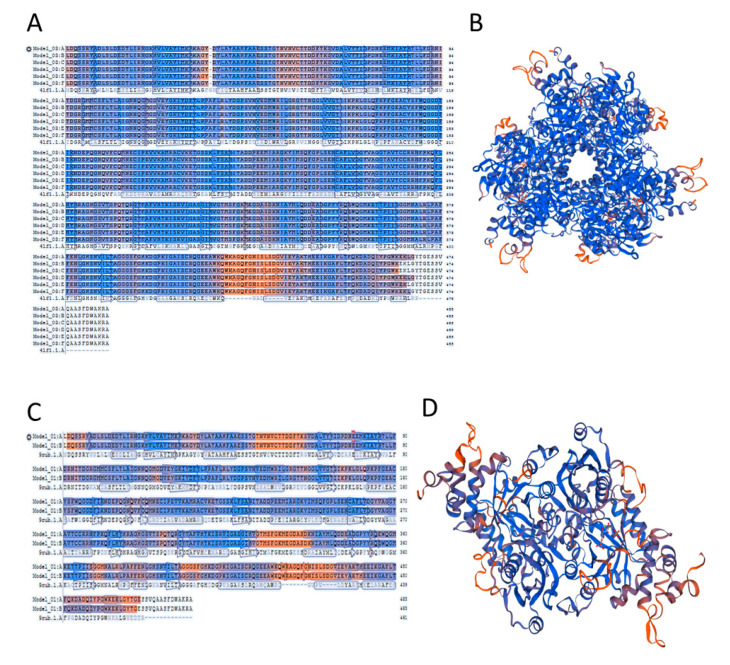
Models’ template alignment (**A**,**C**) and structures (**B**,**D**) of *Symbiodinium* sp. Rubisco based on the 5.hgm.1A structure of form II Rubisco from R. palustris (**A**,**B**) and 9.rub1A from *R. rubrum* (**C**,**D**). Figures generated by SWISS-MODEL. Red colour indicates poorly modelled regions.

**Figure 3 ijms-22-08524-f003:**
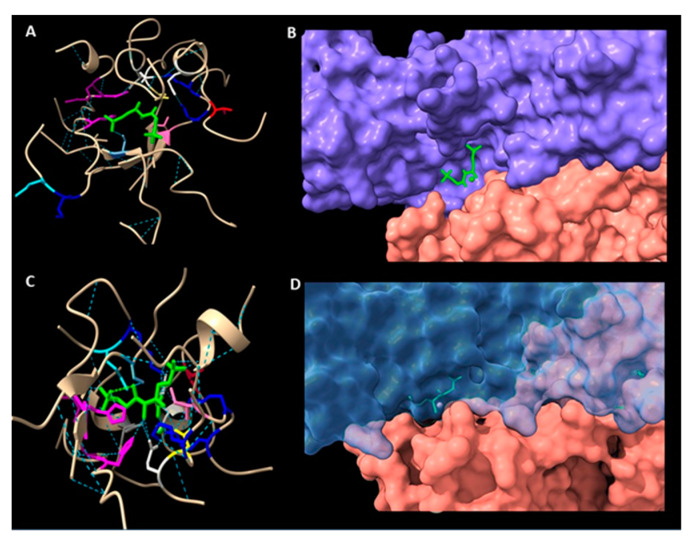
Active site residues arrangement (**A**,**C**) and molecular surface (**B**,**D**) in the *Symbodinium* Rubisco modelled on the *R. rubrum* enzyme (**A**,**B**) and *R. palustris* (**C**,**D**). Amino acids of the active site are coloured identically. Note the presence of a ligand molecule (green sticks), which is RuBP in the *R. rubrum* based structure and CABP in the model based on *R. palustris*.

**Figure 4 ijms-22-08524-f004:**
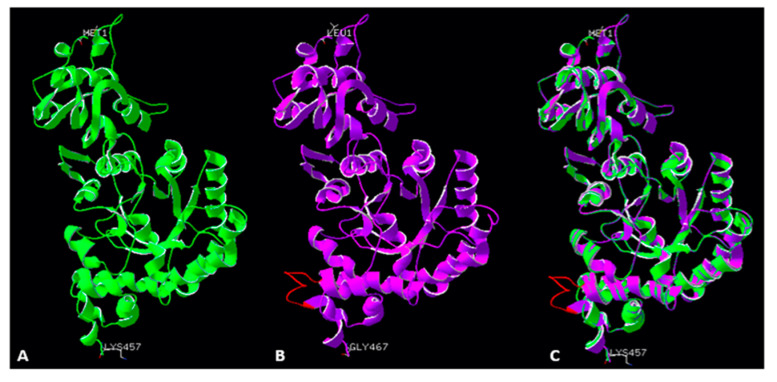
Large subunit monomers from *R. palustris* (**A**), green ribbon structure), modelled *Symbiodinium* sp. Rubisco structure (**B**), violet ribbon structure), and a superimposition of both structures (**C**). Red colour indicates a novel loop (insert 425) in the *Symbiodinium* sp. Rubisco structure.

**Figure 5 ijms-22-08524-f005:**
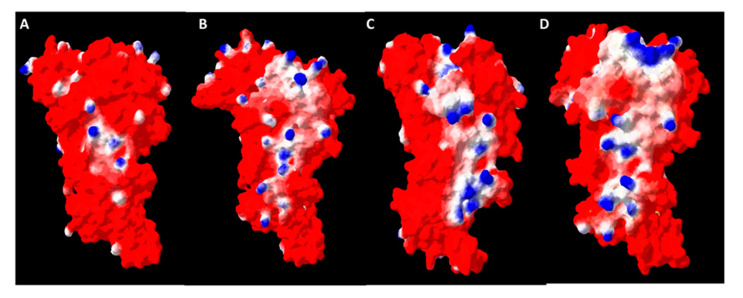
Comparison of the charge distribution over the possible oligomerization interface of *R. rubrum* (**A**), *Symbiodinium* sp. modelled on *R. rubrum* (**B**), *R. palustris* (**C**) and *Symbiodinium* sp. modelled on *R. palustris* (**D**) Rubiscos. Molecular surfaces of protein molecules color-coded by amino acid properties: red—acidic, blue—basic and white—hydrophobic.

**Figure 6 ijms-22-08524-f006:**
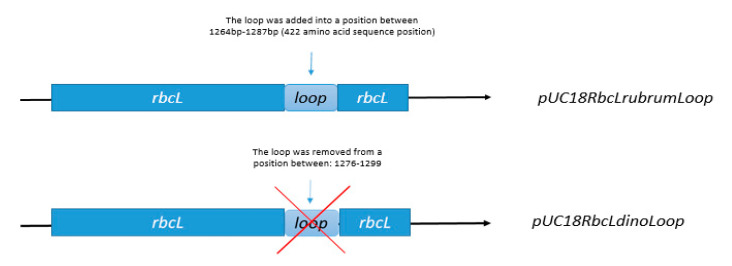
Schematic representation of the genetic constructs used to express mutant RbcL proteins: the 425 insert loop was removed from *rbcL* coding sequence of *Symbiodinium* sp. or added to *rbcL* coding sequence of *R. rubrum*.

**Figure 7 ijms-22-08524-f007:**
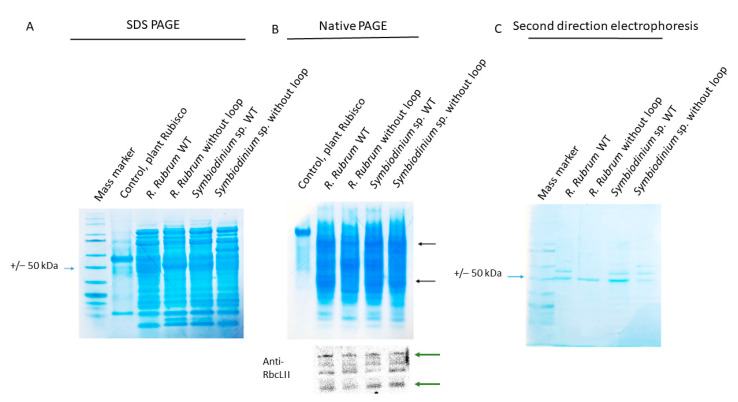
Results of WT *R.rubrum*, WT *Symbiodinium* sp., and the mutated RbcLs (respectively, +loop and −loop) expression in BL21 *E. coli* strain. (**A**) SDS PAGE analysis of the soluble fraction; (**B**) native PAGE of the same fractions and a subsequent Western-blot analysis of the same gel, with anti-form II RbcL antibodies from Agrisera^®^ (AS15 2955) used. Black arrows indicate bands which were analysed on second direction electrophoresis. Green arrows are corresponding to the black ones. (**C**) Second direction electrophoresis in denaturing conditions of the bands indicated by the black arrows of (**B**). All samples in (**B,C**) where standardized for the same total amount of protein.

**Table 1 ijms-22-08524-t001:** Highest scoring homologues of *Rhodospirillum rubrum* Rubisco among dinoflagellates.

Accession Number	Organism	Query Cover [%]	E Value	Percent Identity [%]
Q5ENN5.1	*Heterocapsa triquetra*	97	0.0	67.67
OLP97682.1	*Symbiodinium microadriaticum*	97	0.0	65.95
AAO13031.1	*Prorocentrum minimum*	78	0.0	64.38
Q42813.2	*Lingulodinium polyedra*	97	0.0	64.24

Homologues were searched using the blastP tool with the organism parameter defined to: Dino-flagellates taxid: 2864. Due to the high similarity of sequences between dinoflagellates, only the top 4 are listed in the table. *Symbiodinium microadriaticum* is listed here, as it is the name of an entry; however, in the hereby text we are using simply *Symbiodinium* sp., as it is a convention accepted in most of papers pertaining to dinoflagellates.

**Table 2 ijms-22-08524-t002:** Results for *Symbiodinium* sp. and *R. palustris* Rubisco from the crystallization prediction server XtalPred.

*Organism*	EP Class	RF Class	Length [a.a]	Gravy Index	Instability Index	Isoelectric Point	Coils	Longest Disorder Regions
*Symbiodinium* sp.	2	3	485	−0.33	36.97	5.56	0	4
*R. palustris*	1	3	461	−0.25	38.26	6.28	0	4

EP class (Expert Pool class, score from 1 (best) to 5 (worst)) is a prediction made by combining individual crystallization probabilities calculated for eight protein features into a single crystallization score. Based on this score, the protein is assigned to one of the five crystallization classes. RP (Random Forest Classifier, score from 1 (best) to 10 (worst)) has been extended with other protein features for example surface ruggedness, hydrophobicity, side-chain entropy of surface residues, and based on this score protein is assigned to one of the eleven crystallization classes.

**Table 3 ijms-22-08524-t003:** Templates with highest homology parameters obtained with SWISS-MODEL. The models were built on the templates marked in red.

Organism	ID	Sequence Identity	Sequence Similarity	Coverage	Resolution	Method	GMQE	QSQE
*Riftia pachyptila*	6ius.1.A	68.928	0.515	0.942	2.120	X-ray	0.828	0.79
*Rhodospirillum rubrum*	9rub.1.A	66.377	0.508	0.951	2.600	X-ray	0.820	0.75
*Rhodospirillum rubrum*	2rus.1.A	66.522	0.508	0.948	2.300	X-ray	0.831	0.71
*Rhodopseudomonas palustris*	5hqm.1.A	65.076	0.505	0.951	1.950	X-ray	0.872	0.79
*Rhodopseudomonas palustris*	4lf1.1.A	65.427	0.506	0.942	2.380	X-ray	0.868	0.77
*Gallionella*sp.	5c2c.1.A	63.377	0.495	0.940	2.090	X-ray	0.843	0.75

**Table 4 ijms-22-08524-t004:** Proteins containing fragments homological to the *Symbodinium* sp. RbcL insert in the 425 position.

Protein	Organism	Query Cover [%]	E Value	Percent Identity [%]	Accession Number
Cubilin homolog	*Drosophila willistoni*	100	167	100	XP_023033857.1
Hypothetical protein	*Proteobacteria bacterium*	100	167	100	NDC23151.1
MFS transporter	*Gemmatimonadetes bacterium*	100	167	100	RMH69784.1
Heat shock protein 71 kDa protein	*Fasciola hepatica*	100	168	100	THD27816.1
Response regulator	*Bacteroidetes bacterium*	100	170	100	NQT59094.1

**Table 5 ijms-22-08524-t005:** Sequences of the primers used in mutant’s construction.

Name	Sequence
DinoCLoop-F	GGCGTTATTGAATATGCAAAAACCC
DinoCLoop-R	CTGGCCTGCTTTCCACTGTTTCCATGC
RubrumCLoopF	aggcatggcgcgatggcgtgtttggcaatattagcctgagtgatCCTGTTCTGGATTATGCCCGTGAAC
RubrumCLoopR	CACGCCATCGCGCCATGC

## Data Availability

All data are presented within manuscript and Appendix A.

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
