# Peer review of "Insights into the Structure of Rubisco from Dinoflagellates-In Silico Studies"

_ijms, 2021, doi:10.3390/ijms22168524_

Round 1
Reviewer 1 Report
The article by Małgorzata Rydzy and co-authors "Insights into the structure of Rubisco from Dinoflagellates -in silico studies" is devoted to the construction of dinoflagellates (Symbiodinium sp.) Rubisco enzyme model using in silico tools.
The hypothesis of the study is based on a serious assumption about the homology of structures of different types of Rubisco. However, it should be borne in mind that this is the only approach that the authors could have taken, since there is not enough experimental data on the structure of dinoflagellates Rubisco.
As I moved along the text of the article, I had questions, however, they were resolved by further descriptions or obtained results.
The work was done at a very high methodological and semantic level. The remarks are of a "cosmetic" nature.
The share of sources older than 5 years is high. I am sure there will be a sufficient number of publications released over the past few years, which will confirm the ideas and assumptions of the authors.
Sections of the chapter materials and methods (3.5. - 3.7.) аre not accompanied by references. If these are not the author's methods (although there are no indications of this), these sections should be accompanied with appropriate sources.
The conclusion largely repeats obtained results. I would recommend the authors to develop the thought postulated at the end of the conclusion. Describe in more detail how their model (and general progress in understanding the workings of Rubisco dinoflagellates) can be used in perspective.
Author Response
Please, see the attachement

Reviewer 2 Report
Despite the reported topic is of some interest, I would first request the authors to rewrite the entire manuscript draft. It is absolutely necessary sending it for professional English editing/native speaker correction before it can be further considered. In its current form the paper draft is almost not understandable in most of its parts.
Author Response
Please see the attachement.

Reviewer 3 Report
The manuscript by Małgorzata Rydzy anc co-workers analyzed Rubisco the best-studied enzyme involved in the dark phase of photosynthesis. The authors using in silico tools to generate the possible structure of Rubisco from dinoflagellate representative, Symbiodinium sp. not yet resolved.
From in-silico structural data obtained by using a 3D structures template from Rhodospirillum rubrum and Rhodopseudomonas palustris the authors point out the probably hexameric nature of Symbiodinium sp. Rubisco. Moreover, the authors, individuated a novel motif, consisting of a helix extension and a loop not directly involved in Rubisco's catalytical reactions but probably involved in regulatory function.
The manuscript deals an important research subject, the efforts in the understanting of structure of Rubisco will be useful to improve its catalytic activity and consequently improving of algae/plant productivity.
The experiments presented in this work are quite good conducted and are useful for the elucidation of issues presented in the paper and they are enough to draw conclusions.
I have only a techincal suggestion for the authors in order to overcome the ab-problems. Could be possible to perform a Trypsin-in gel digestion and LC-MS-MS identification, in order to assesed the identity of each band?
In general this paper can contribuite and open new questions about the Rubisco structural studies, the next step should be the isolation of native complexes in order to enstablish, by cryo-EM, the quaternary conformation.
Above i listed some small mistakes:
Line 16: space between genre/species
Line 68: space between genre/specie
Line 100: space “also in silico”
Line 132: space between “rubrum template”
Line 140: space between “to Euglena’s”
Line 154: space “the higher”
Line 156: space "their number"
Author Response
We are grateful to the Reviewer for reading and critical evaluation of our manuscript.
The formatting mistakes (lacking spaces) have been corrected. We noticed that such a problem occurs after placing the text into the IJMS template, and we hope it will no reappear after submitting the file now.
In regard to the question about the identification of proteins in gel-cut bands, in general, we agree with the Reviewer. Actually, we have considered MS/MS. However, However, our previous expertise with this method shows, that it only confirms the presence of protein (which is already done by antibodies), but very rarely allow to have full protein sequence coverage. Therefore, we cannot clearly say if the bands are shorter peptides due to proteolytic digestion or some misleads during translation/folding. For that reason, we believe there is no point right now to conduct that experiments.